# Characterization of patients with advanced chronic pancreatitis using natural language processing of radiology reports

**Fagen Xie[1], Qiaoling Chen[1], Yichen Zhou[1], Wansu Chen[1], Jemianne Bautista[2], Emilie T. Nguyen[2], Rex A. Parker[2], Bechien U. Wu [3]***

**1** Research & Evaluation, Kaiser Permanente Southern California, Pasadena, California, United States of America, **2** Department of Radiology, Kaiser Permanente Los Angeles Medical Center, Los Angeles, California, United States of America, **3** Center for Pancreatic Care, Division of Gastroenterology, Kaiser Permanente Los Angeles Medical Center, Los Angeles, California, United States of America

* Bechien.u.wu@kp.org

## Abstract

### Study aim

To develop and apply a natural language processing algorithm for characterization of patients diagnosed with chronic pancreatitis in a diverse integrated U.S. healthcare system.

### Methods

Retrospective cohort study including patients initially diagnosed with chronic pancreatitis (CP) within a regional integrated healthcare system between January 1, 2006 and December 31, 2015. Imaging reports from these patients were extracted from the electronic medical record system and split into training, validation and implementation datasets. A natural language processing (NLP) algorithm was first developed through the training dataset to identify specific features (atrophy, calcification, pseudocyst, cyst and main duct dilatation) from free-text radiology reports. The validation dataset was applied to validate the performance by comparing against the manual chart review. The developed algorithm was then applied to the implementation dataset. We classified patients with calcification(s) or ≥2 radiographic features as advanced CP. We compared etiology, comorbid conditions, treatment parameters as well as survival between advanced CP and others diagnosed during the study period.

### Results

6,346 patients were diagnosed with CP during the study period with 58,085 radiology studies performed. For individual features, NLP yielded sensitivity from 88.7% to 95.3%, specificity from 98.2% to 100.0%. A total of 3,672 patients met cohort inclusion criteria: 1,330 (36.2%) had evidence of advanced CP. Patients with advanced CP had increased frequency of smoking (57.8% vs. 43.0%), diabetes (47.6% vs. 35.9%) and underweight body mass index (6.6% vs. 3.6%), all p<0.001. Mortality from pancreatic cancer was higher in advanced CP (15.3/1,000 person-year vs. 2.8/1,000, p<0.001). Underweight BMI (HR 1.6,

**Data Availability Statement:** The Kaiser Permanente Southern California institutional policy requires a data transfer agreement be executed between KPSC and the individual recipient entity

prior to transmittal of patient-level data outside KPSC. This is a legal requirement. Requests for data can be addressed to the Central Business Office of the Department of Research and Evaluation (contact via Judy.J.Angmorter@kp.org).

**Funding:** BW, U01 DK 108314, National Institutes of Diabetes, Digestive and Kidney Diseases, https://www.niddk.nih.gov/. The funders had no role in study design, data collection and analysis, decision to publish or preparation of the manuscript.

**Competing interests:** The authors have declared that no competing interests exist.

95% CL 1.2, 2.1), smoking (HR 1.4, 95% CL 1.1, 1.7) and diabetes (HR 1.4, 95% CL 1.2, 1.6) were independent risk factors for mortality.

## Conclusion

Patients with advanced CP experienced increased disease-related complications and pancreatic cancer-related mortality. Excess all-cause mortality was driven primarily by potentially modifiable risk factors including malnutrition, smoking and diabetes.

## Introduction

Chronic pancreatitis is a distinct pathologic and clinical entity along the spectrum of inflammatory conditions that involve the pancreas. Previous estimates indicate a population prevalence ranging from 50/100,000 persons [1] to 91.9/100,000 persons [2] in the United States. However, reliable real-world data relating to the natural history of chronic pancreatitis is limited based on the inability to accurately characterize patients diagnosed in the context of routine clinical practice. In particular, given the complexity of establishing a diagnosis of chronic pancreatitis [3], previous studies evaluating accuracy of diagnosis codes alone have found accuracy rates below 50% [4].

A key challenge in studying chronic pancreatitis at the population level is the lack of a structured format for reporting of pancreatitis-related imaging findings. A systematic approach to accurately identifying these features from the free-text of existing radiology reports is a major step towards enhancing our ability to study the natural history of chronic pancreatitis (CP) on a large-scale. The application of clinical natural language processing (NLP) has the potential to address these challenges. By developing various methods for semantic processing and analysis of clinical texts, these methods can be applied to a variety of clinical applications [5–11]. To date, this technology has not been applied to characterize features specific to chronic pancreatitis.

The objective of this study was to leverage NLP technologies to perform a systematic assessment of patients diagnosed with CP in a diverse, integrated community-based healthcare system. Specifically, we sought to identify a subset of patients with advanced CP with respect to both radiographic findings and clinical presentation.

## Methods

### Study setting and patient population

This study was approved by the Kaiser Permanente Southern California Institutional Review Board, protocol #11121. Waiver of informed consent was granted due to the data-only nature of study (no direct patient contact). We conducted a retrospective cohort study to characterize patients with advanced features of chronic pancreatitis that were diagnosed in a large racially/ethnically diverse community-based population. The study was conducted among patients with an initial diagnosis of chronic pancreatitis (International Classification of Disease 9th revision 577.1 or 10th revision K86.1) from Kaiser Permanente Southern California (KPSC) between January 2006 and December 2015. KPSC is an integrated healthcare delivery system composed of 15 hospitals and more than 220 satellite medical offices throughout southern California with a comprehensive electronic medical record system, providing comprehensive care for over 4.6 million active members [12]. The electronic medical record (EMR) systems

captures members' care information including structured as well as unstructured data, such as radiology reports and clinical notes.

Patients with a history of pancreatic cancer prior to diagnosis of CP, those with a diagnosis of CP prior to 2006 and those with less than one year of continuous health plan enrollment prior to the diagnosis of CP were excluded. In addition, all patients were required to have at least 1 pancreas-related image (abdominal ultrasound, computed tomography CT or magnetic resonance imaging MRI). The present study was approved by the KPSC Institutional Review Board.

## Natural language processing for characterization of pancreatic imaging findings

**Feature terms.** The recent American Pancreatic Association (APA) clinical guideline established a number of pancreas imaging features to characterize CP [3]. The following five features were included in the present study: atrophy, calcification, pseudocyst, cyst and ductal dilatation. The search keywords or terms for each feature were compiled based on the APA clinical guideline definitions, ontologies in the Unified Medical Language System [13] and enriched from training datasets during the algorithm development to capture additional possible linguistic variations. For example, the compiled terms for atrophy included atrophy, atrophic and atrophied. In addition to the search terms, excluded terms were also identified for pseudo cyst and duct dilation features to exclude the report for processing. The excluded terms of "pseudo cyst", "pseudo-cyst" and "cystic duct" are applied for identifying the feature of cyst because "pseudocyst" and "cyst" were identified independently and exclusively while the term of "common duct", "bile duct" or "pancreatic duct side branch" is excluded for ductal dilation because the study aimed to identify main pancreatic ductal dilation. Furthermore, a modifier term of "duct" or "ductal" was used to define the feature of ductal dilation. Specifically, the feature of ductal dilatation is identified by searching a feature term (i.e., "dilation") and a modifier term (i.e., "duct") within ten words. The compiled keywords and their corresponding modifier terms or exclusion terms are summarized in Table 1 in S1 Table.

**Imaging reports.** The imaging reports (CT, MRI and ultrasound) for patients diagnosed with chronic pancreatitis during the study period were first extracted from the KPSC EMR system. The extracted imaging reports are free-text format, and most of them contain the sections of clinical history, procedure, technique, finding. Imaging reports without the string "pancreas" were removed. The remaining imaging reports were used to form the training, validation and implementation datasets (see below).

**Training dataset.** A sample of 100 CP patients (20 for each imaging feature) was randomly selected from the study cohort. Their corresponding radiology imaging reports (total = 1,253) were manually reviewed by the clinical study team to determine the presence of each of the individual pancreatitis-related imaging finding. The results were used for initial algorithm development for each imaging feature. An additional set of 500 imaging reports from a separate sample of 450 patients were then randomly selected from the remaining imaging reports and reviewed manually by the study radiologists. The results of the manual review of these 500 imaging reports were used to provide further refining of the rule-based computer-generated algorithms.

**Validation dataset.** A total of randomly selected 500 imaging reports from another sample of 453 patients were used to generate the reference standard for the purposes of validation. All reports were fully reviewed by the study radiologists to identify the study interested five imaging features. Implementation dataset: The final computer algorithm was implemented among the remaining imaging reports (exclusions of training and validation reports). The

results of the algorithm-based classification were then used to determine the pancreatic imaging features among patients included in the CP study cohort.

**Imaging report preprocessing.** The extracted imaging reports were first pre-processed. This included cleaning special characters, spelling checking and correction for these mistyped, misspelled or concatenated words detected from our development training datasets, section detection, sentence separation, and tokenization (i.e., segmenting text into linguistic units such as words and punctuation). For example, the phrase "cyst lesion" was mistakenly concatenated as "cystlesion" and "suspicious" was mistyped as "supsicious" in some reports.

**NLP algorithm development.** An internally developed NLP platform installed on a Linux server was used to develop the algorithm. Built by using Python programming language, the platform integrated a number of well-known open source application programming interfaces including the Negex and ContextNLP [5, 6], the Natural Language Toolkit (NLTK) [14] and Stanford Core NLP [15]. A rule-based NLP algorithm was developed for each imaging feature using the two training datasets as previously mentioned. The rules were first defined at the sentence level and the results were later combined to reflect report or patient level information. The detailed steps for algorithm development and validation are presented in S1 Appendix.

**NLP algorithm validation.** The manual chart review results of the validation dataset served as the reference standard to evaluate the performance of the computerized algorithm at imaging report level. Any discrepancy between manual review and NLP algorithm results were fully adjudicated by the clinical team (JB, EN, BW).

**NLP algorithm implementation.** The validated NLP algorithm was applied to the implementation dataset to classify the five pancreatic imaging features for patients included in the study cohort.

## Data analysis

To qualify as advanced CP, patients were required to either have calcification(s) on imaging or at least 2 of the 5 previously delineated abnormal imaging findings. Patients with presence of only cyst and/or pseudocyst were not classified as advanced CP. To characterize differences in the clinical profile of patients with advanced CP we performed Chi-square test, t-test or Wilcoxon test to assess the age, sex and racial/ethnic distribution as well as the frequency distribution of etiologic risk factors (alcohol, smoking), comorbid illnesses (diabetes, underweight body habitus: body mass index<18.5) and pancreatic exocrine insufficiency (fecal elastase <200 g/dL, prescription for pancreatic enzyme replacement therapy). We also identified patients in the study cohort with chronic opioid use defined as history of opioid dispensations in the outpatient setting for duration >6 months at any time during follow-up.

To evaluate differences in the natural history of patients with advanced imaging features compared to others diagnosed with CP during the study period we performed survival analysis using the Kaplan-Meier method to assess rates of all-cause mortality. We further compared rates of pancreatic cancer and pancreatic cancer-related mortality based on CP status. Finally, we performed multivariable Cox proportional hazards regression to evaluate independent risk factors for all-cause mortality across the study cohort.

## Results

### Chronic pancreatitis study cohort

We identified a total of 7,072 patients diagnosed with chronic pancreatitis based on ICD-9 code during the study period. Among these, 6,346 (89.7%) had pancreas imaging obtained. Following study inclusion and exclusion criteria, a total of 3,672 patients were included in the final study cohort (see Fig 1 for cohort assembly). The median age at time of CP diagnosis was

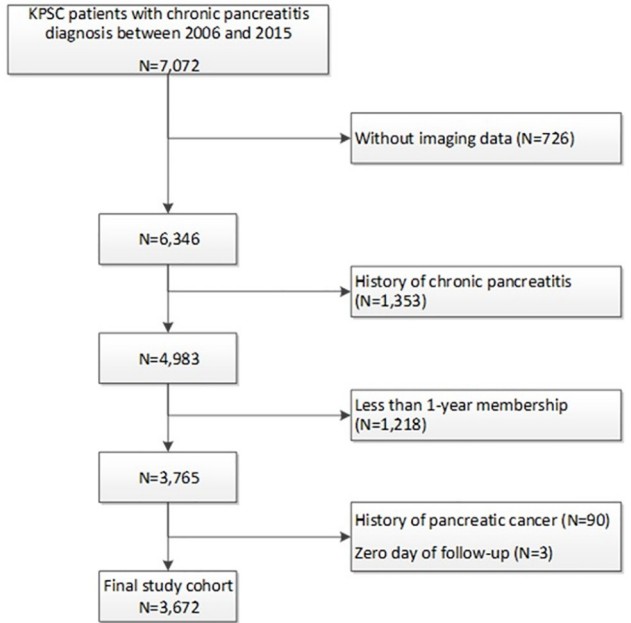

**Fig 1. Flow diagram for cohort assembly.**

58.0 years (interquartile range 46.0, 70.0). Overall, 1794 (48.9%) of the study cohort were women and 1,813 (49.4%) were non-Hispanic White, 1,037 (28.2%) Hispanic, 517 (14.1%) Black and 233 (6.3%) were Asian.

## Natural language processing for pancreatitis-imaging findings

A total of 58,085 radiology imaging reports incorporating the pancreas-related keywords were retrieved among 6,346 eligible CP patients. A total of 726 (~10%) CP patients did not have any imaging reports that contained the pancreas-related keyword and thus were excluded. The mean number of imaging reports per patient was 9.1 (standard deviation 9.6). The results of the algorithm-based identification for the five pancreatic features applied to the implementation dataset at the imaging report level and patient level are shown in Table 2 in S1 Table. The definite (positive) rate of these features ranged from 5.0% (atrophy) to 12.29% (calcification) at report level and 19.16% (pseudocyst) to 27.33% (calcification) at patient level. The NLP algorithm also identified a certain percentage of probable or likely cases for each feature, which ranged from 0.13% (atrophy) to 2.47% (pseudocyst) at the report level and from 0.3% (atrophy) to 4.83% (pseudocyst) at the patient level. Table 3 in S1 Table summarizes the distribution of the total number of definite or probable features at the report level as well as at the patient level. Overall, nearly 33.0% of imaging reports and 56.0% of CP patients presented with at least one of the five common pancreatic imaging features.

Data on the comparison of NLP algorithm results versus the manual results for each imaging feature from the validation set are shown in Table 1. Compared to manual validation, the NLP algorithm demonstrated sensitivity ranging from 88.7% (cyst) to 95.3% (atrophy), specificity from 98.2% (cyst) to 100.0% (ductal dilation), PPV from 85.5% (cyst) to 100.0% (ductal dilation), NPV from to 98.7% (cyst) to 99.6% (atrophy), F-score from 0.87 (cyst) to 0.97 (pseudocyst, ductal dilation). When "Definite" and "Probable" were combined as one category, the sensitivity of atrophy and ductal dilation remained same while the sensitivity was increased to

**Table 1. Accuracy of the computerized algorithm based on the validation dataset.**

| Pancreatic features | TP | TN | FN | FP | Sensitivity (%) | Specificity (%) | PPV (%) | NPV (%) | F-score |
|---|---|---|---|---|---|---|---|---|---|
| Definite and probable as separate groups | | | | | | | | | |
| Atrophy | 41 | 456 | 2 | 1 | 95.3 | 99.8 | 97.6 | 99.6 | 0.96 |
| Calcification | 77 | 416 | 4 | 3 | 95.1 | 99.3 | 96.3 | 99.0 | 0.95 |
| Pseudocyst | 75 | 421 | 3 | 1 | 95.2 | 99.8 | 98.7 | 99.3 | 0.97 |
| Cyst | 47 | 439 | 6 | 8 | 88.7 | 98.2 | 85.5 | 98.7 | 0.87 |
| Ductal dilatation | 83 | 412 | 5 | 0 | 94.3 | 100.0 | 100.0 | 98.8 | 0.97 |
| Definite and probable as one single group | | | | | | | | | |
| Atrophy | 41 | 456 | 2 | 1 | 95.3 | 99.8 | 97.6 | 99.6 | 0.96 |
| Calcification | 79 | 416 | 3 | 2 | 96.3 | 99.5 | 97.5 | 99.3 | 0.97 |
| Pseudocyst | 77 | 421 | 1 | 1 | 98.7 | 99.8 | 98.7 | 99.8 | 0.99 |
| Cyst | 52 | 439 | 5 | 4 | 91.2 | 98.9 | 92.9 | 99.1 | 0.92 |
| Ductal dilatation | 83 | 412 | 5 | 0 | 94.3 | 100.0 | 100.0 | 98.8 | 0.97 |

FN: false negative; FP: false positive; TN: true negative; TP: true positive

PPV: positive predictive value; NPV: negative predictive value

91.2%, 96.3%, 98.7% for cyst, calcification, pseudocyst, respectively and PPV was increased to 92.9% for cyst, and 97.5% for calcification. The F-score of cyst became 0.92. Details of error analysis between NLP and chart review are presented in S1 Error analysis.

A total of 1,330 (36.2%) patients had evidence of calcification and/or at least 2 imaging abnormalities and qualified as advanced CP as defined by the present study. Baseline demographic and clinical characteristics for the study cohort stratified by CP status are presented in Table 2. Patients with advanced CP tended to be older at the time of diagnosis (median age 63 years vs 54 years, p<0.001) and were more frequently male (55.8% vs 48.5% p<0.001) as well as White race/ethnicity (56.9% vs 45.1%, p<0.001). Smoking (57.8% vs. 43.0%, p<0.001) and alcohol (31.7% vs 28.6%, p<0.001) were also more common among patients with features of advanced CP.

**Disease-associated morbidity.** A total of 1,137 (31%) of patients received opioids for >6 months during the study period. Chronic opioid use was more common among patients with advanced CP (35.1% vs 28.6%, p<0.001). Diabetes (47.6% vs. 35.9%, p<0.001) as well as underweight body mass index (6.6% vs. 3.6%, p<0.001) at the time of diagnosis were also more common among patients with advanced imaging features of CP.

**Mortality.** A total of 928 (25.3%) patients in the cohort died during the study period (mortality rate 62/1,000 person-year). Among patients with advanced CP, the mortality rate was 83/1,000 person-year. A kaplan-meier survival plot stratified by disease category is presented in Fig 2. Advanced CP was associated with worsened mortality (median 5-year mortality 33.2% (95% CI 30.3, 36.1) vs 21.8% (19.9, 23.8), p < .001). The most frequent causes of mortality across the study groups are listed in Table 3. Frequency of pancreatic cancer-related death was increased among patients with advanced CP (5.9% vs. 1.2%, p < .001) with pancreatic cancer-related mortality of 15/1,000 person-years among patients with advanced CP. The results of multivariable Cox proportional hazards analysis are presented in Table 4. In multivariable analysis, advanced CP status was not independently associated with increased risk of mortality whereas underweight BMI (hazard ratio HR 1.57, 95% CL 1.20, 2.05), active smoking (HR 1.36, 95% CL 1.10, 1.68) and diabetes (HR 1.35, 95% CL 1.18, 1.55) at diagnosis were each independent risk factors for all-cause mortality.

**Table 2. Baseline and follow-up characteristics of KPSC patients with chronic pancreatitis between 2006 and 2015.**

| | Advanced CP (N = 1,330) | Not advanced AP (N = 2,342) | P-value |
|---|---|---|---|
| **Patient Baseline Demographics** | | | |
| **Age at CP diagnosis** | | | <.001 |
| Mean (SD) | 62.8 (15.34) | 53.7 (17.42) | |
| Median (Q1, Q3) | 63 (53.0, 74.0) | 54 (42.0, 67.0) | |
| **Gender** | | | <.001 |
| Female | 588 (44.2%) | 1206 (51.5%) | |
| Male | 742 (55.8%) | 1136 (48.5%) | |
| **Race/Ethnicity** | | | <.001 |
| Asian | 74 (5.6%) | 159 (6.8%) | |
| Black | 187 (14.1%) | 330 (14.1%) | |
| Hispanic | 294 (22.1%) | 743 (31.7%) | |
| White | 757 (56.9%) | 1056 (45.1%) | |
| Others/unknown | 18 (1.4%) | 54 (2.3%) | |
| **Median household income** | | | 0.094 |
| < = $45,000 | 298 (22.4%) | 581 (24.8%) | |
| $45,001-$80,000 | 570 (42.9%) | 987 (42.1%) | |
| $80,001+ | 348 (26.2%) | 545 (23.3%) | |
| Unknown | 114 (8.6%) | 229 (9.8%) | |
| **Membership duration: pre-index CP diagnosis (year)** | | | <.001 |
| Mean (SD) | 16.0 (12.78) | 13.1 (11.74) | |
| Median (Q1, Q3) | 12.6 (5.0, 24.0) | 9.4 (4.1, 18.9) | |
| **Patient baseline clinical characteristics** | | | |
| **Pancreatic surgery** | 53 (4%) | 41 (1.8%) | <.001 |
| **Alcohol** | | | 0.008 |
| No | 625 (47%) | 1074 (45.9%) | |
| Yes | 422 (31.7%) | 669 (28.6%) | |
| Unknown | 283 (21.3%) | 599 (25.6%) | |
| **Smoking status** | | | <.001 |
| Non-smoker | 443 (33.3%) | 1055 (45%) | |
| Former smoker | 482 (36.2%) | 648 (27.7%) | |
| Current smoker | 287 (21.6%) | 358 (15.3%) | |
| Unknown | 118 (8.9%) | 281 (12%) | |
| **Diabetes** | 633 (47.6%) | 840 (35.9%) | <.001 |
| **Acute pancreatitis** | 439 (33%) | 809 (34.5%) | 0.345 |
| **BMI**, Mean (SD) | 25.8 (5.58) | 28.3 (6.66) | <.001 |
| Under Weight | 88 (6.6%) | 84 (3.6%) | <.001 |
| Normal Weight | 562 (42.3%) | 687 (29.3%) | |
| Over weight | 403 (30.3%) | 756 (32.3%) | |
| Obese | 272 (20.5%) | 792 (33.8%) | |
| Unknown | 5 (0.4%) | 23 (1%) | |
| **Follow-up** | | | |
| **Length of follow-up (year)** | | | 0.011 |
| Mean (SD) | 3.9 (2.94) | 4.2 (3.13) | |
| Median (Q1, Q3) | 3.2 (1.4, 5.9) | 3.5 (1.5, 6.5) | |
| **Pancreatic cancer** | 97 (7.3%) | 43 (1.8%) | <.001 |
| **Chronic opioid use** | 467 (35.1%) | 670 (28.6%) | <.001 |
| **Pancreatic enzyme insufficiency***  | 569 (42.8%) | 538 (23%) | <.001 |

(*Continued*)

**Table 2.** (Continued)

|  | Advanced CP (N = 1,330) | Not advanced AP (N = 2,342) | P-value |
|---|---|---|---|
| Pancreatic enzyme supplementation | 558 (42%) | 536 (22.9%) | <.001 |
| Fecal elastase <200 mg/dL | 79 (5.9%) | 21 (0.9%) | <.001 |
| **5-year Mortality, %(95%CI)** | 33.2 (30.3, 36.1) | 21.8 (19.9, 23.8) | <.001 |

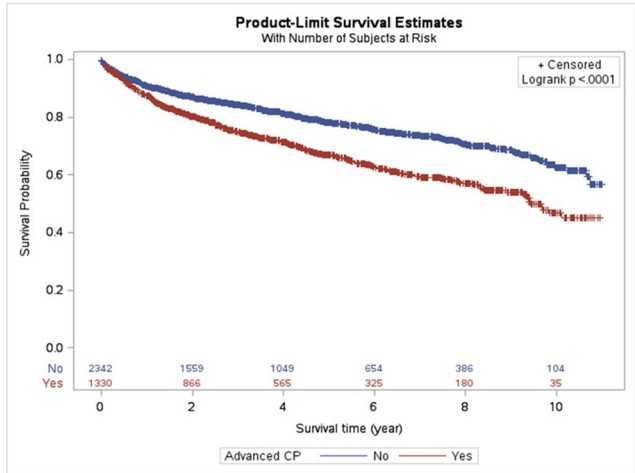

**Fig 2. Survival analysis, Kaplan-Meier curve for all-cause mortality.** Advanced chronic pancreatitis based on radiographic findings abstracted through natural language processing.

## Discussion

We developed a natural language processing algorithm to identify radiographic features commonly associated with chronic pancreatitis from the free text of radiology reports. This algorithm was then applied on existing data to help characterize a cohort of patients diagnosed with chronic pancreatitis in a large integrated healthcare system. Through use of the NLP algorithm we were able to identify a subset of patients with advanced features of chronic pancreatitis based on radiographic findings that corresponded to increased rates of diabetes, chronic

**Table 3. Causes of mortality.**

|  | Advanced CP (N = 1,330) | Not advanced CP (N = 2,342) | Total (N = 3,672) |
|---|---|---|---|
| **Person-years** | 5174.6 | 9737.8 | 14913 |
| **Death from all causes** | 432 (32.5%) | 496 (21.2%) | 928 (25.3%) |
| Pancreatic cancer | 79 (5.9%) | 27 (1.2%) | 113 (3.1%) |
| Other malignancies | 94 (7.1%) | 94 (4%) | 211 (5.7%) |
| Circulatory system disease | 80 (6%) | 106 (4.5%) | 214 (5.8%) |
| Diseases of the digestive system | 61 (4.6%) | 82 (3.5%) | 149 (4.1%) |
| Diseases of the respiratory system | 25 (1.9%) | 24 (1%) | 61 (1.7%) |
| Endocrine, nutritional and metabolic diseases | 28 (2.1%) | 34 (1.5%) | 73 (2%) |
| Other diagnosis | 52 (3.9%) | 111 (4.7%) | 183 (5%) |
| Unknown | 13 (1%) | 18 (0.8%) | 31 (0.8%) |

**Table 4. Adjusted hazard ratios for mortality.**

| | Mortality | | |
| --- | --- | --- | --- |
| | Hazard Ratio | 95% Confidence Limits | | P-value |
| Advanced CP (ref = no) | 1.11 | 0.97 | 1.27 | 0.14 |
| Age | 1.04 | 1.03 | 1.04 | <.001 |
| Gender (male vs. female) | 1.16 | 1.01 | 1.32 | 0.03 |
| Race/ethnicity (Ref = White) | | | | 0.02 |
| Asian | 0.75 | 0.56 | 1.00 | 0.05 |
| Black | 0.99 | 0.81 | 1.20 | 0.89 |
| Hispanic | 0.83 | 0.70 | 0.99 | 0.04 |
| Others/unknown | 1.62 | 1.01 | 2.60 | 0.05 |
| Median household income (ref: < = $45,000) | | | | 0.15 |
| $45,001-$80,000 | 0.99 | 0.84 | 1.16 | 0.86 |
| $80,001+ | 0.84 | 0.70 | 1.02 | 0.08 |
| Unknown | 0.82 | 0.62 | 1.09 | 0.17 |
| BMI (ref = normal weight) | | | | <.001 |
| Underweight | 1.57 | 1.20 | 2.05 | 0.001 |
| Over weight | 0.72 | 0.61 | 0.84 | <.001 |
| Obese | 0.67 | 0.56 | 0.80 | <.001 |
| Unknown | 6.00 | 3.49 | 10.30 | <.001 |
| Alcohol (Ref = No) | | | | 0.99 |
| Unknown | 0.99 | 0.81 | 1.21 | 0.88 |
| Yes | 1.00 | 0.84 | 1.18 | 0.97 |
| Smoking (Ref = Non-smoker) | | | | 0.001 |
| Current smoker | 1.36 | 1.10 | 1.68 | 0.004 |
| Former smoker | 1.27 | 1.08 | 1.50 | 0.004 |
| Unknown | 1.51 | 1.17 | 1.95 | 0.002 |
| Acute pancreatitis (Yes vs. no) | 1.09 | 0.94 | 1.26 | 0.26 |
| Diabetes (Yes vs. no) | 1.35 | 1.18 | 1.55 | <.001 |

opioid use, underweight body mass index as well as pancreas cancer-related mortality. In multivariable analysis, smoking, diabetes and underweight body status were independent risk factors for all-cause mortality.

The ability to further characterize patients diagnosed in a real-world setting is an important step to improving understanding of the natural history of chronic pancreatitis. A major limitation to studying chronic pancreatitis at the population-level has been the limited accuracy of diagnosis codes with positive-predictive value <50% compared to manual chart review using established clinical criteria [4]. As radiographic imaging features are one of the key objective criteria used in the diagnosis and staging of chronic pancreatitis, we sought to develop a rule-based NLP algorithm to identify five of the most common pancreatic imaging features from radiology reports associated with chronic pancreatitis. Compared with findings from the manual review, the NLP algorithm produced a high level of performance for each of the specific pancreatitis-related features.

Findings from the present study with respect to etiologic risk factors including increased frequency of smoking and alcohol among patients with more advanced radiographic findings are consistent with previous prospective studies of chronic pancreatitis from North America [16–18]. It is also interesting to note that although the base cohort was racially and ethnically diverse, patients with advanced findings were disproportionately white and male.

In terms of disease management, the proportion of patients with advanced CP that received testing or treatment for exocrine insufficiency was relatively low (40%). This is likely the result of broad uncertainty regarding appropriate testing and treatment for exocrine insufficiency [19]. Moreover, previous literature indicates pervasive issues regarding under-treatment for this aspect of chronic pancreatitis including a survey from the Netherlands that indicated >70% of patients with advanced CP continued to experience symptoms of steatorrhea despite enzyme replacement [20]. These findings highlight an important opportunity to improve care for these patients that may help address the increased prevalence of underweight body mass index and worsened survival in this patient population.

In the present study, 35% of patients with advanced CP were treated with chronic opioids (35%). This is lower than previous reports from the North American Pancreatitis Study-2 where nearly half (47%) of patients indicated chronic opioid use based on self-report [21]. This discrepancy could be due to differences in care-setting given previous reports were conducted primarily based on data from tertiary care referral centers as well as recent initiatives to curb use of these medications within the KPSC health system [22].

Increased mortality among patients with chronic pancreatitis has been previously described [23,24]. However, all-cause mortality among the advanced CP patients in the present study (83/1,000 pyr) was notably higher than that reported from previous longitudinal studies of CP (25-77/1,000 pyr) [24–26]. In addition, the rate of pancreatic cancer-related mortality (15/1,000 pyr) was strikingly higher than the rate (0.06/1,000 pyr) reported in a recent large population-based study in the Netherlands[24]. We believe that discordant findings between the present study and previous examinations are likely related to differences in design such that previous endoscopy-based cohorts [25] may not have captured the full spectrum of patients with advanced chronic pancreatitis whereas population-based estimates relied upon either diagnosis codes exclusively [24] or included a relatively small number of carefully curated CP cases [23, 26].

There were several limitations to the current study. First, there were limitations with respect to the NLP algorithm such that features were abstracted from the free text of radiology reports. Therefore, it is likely that unreported or infrequent findings would not be adequately captured. This was the rationale to limit the algorithm to identify the five most common findings associated with chronic pancreatitis. Second, based on the retrospective nature of the study we were unable to capture direct measures of disease manifestations such as chronic abdominal pain or steatorrhea. As a result, the present analyses were limited to measures of disease treatment including opioid dispensation(s) as well as pancreatic enzyme supplementation.

Despite these limitations, findings from the present study have several important implications for both further research as well as clinical practice. From a research perspective, the ability to apply an automated NLP-based algorithm to identify features related to chronic pancreatitis from the free-text of radiology reports at-scale offers a tremendous opportunity to gain further insight into the natural history of disease as illustrated by several of the present analyses. From a clinical perspective, it is important to note that although patients with advanced CP experienced greater overall mortality, the results from multivariable analysis suggest that this finding is driven primarily through potentially modifiable risk factors including diabetes, smoking and underweight BMI. This stresses the importance of adequately managing these comorbid conditions in order to improve long-term survival particularly among patients with advanced stages of chronic pancreatitis. Finally, the relatively high rate of pancreatic cancer-related mortality among patients with advanced features of CP suggests that this is a population that may benefit from further strategies for early detection of pancreatic cancer.

It should also be noted out that the computerized algorithm developed in the present study can be applied in other healthcare systems as well as potentially adapted for other disease

states. It may yield some varied results due to the variation in format and presentation of clinical reports. However, the accuracy should not be significantly affected because the algorithm was not targeted or limited to any fixed/strict formatted reports. In addition, our approach and process can be modified to address other medical conditions with some modifications, such as replacing the keywords or terms of interest, corresponding modifiers and potential excluded terms, etc.

In summary, we have developed and applied a natural-language processing algorithm to identify features commonly associated with chronic pancreatitis from the free text of radiology reports. Using this algorithm, we were able to identify a subset of patients with advanced disease among those diagnosed with chronic pancreatitis in a large integrated healthcare system. Patients with advanced chronic pancreatitis were at increased risk for pancreatitis-associated morbidity as well as pancreatic cancer-related mortality. Excess overall mortality observed among patients with advanced CP was driven primarily by underweight BMI, smoking and diabetes. Greater emphasis on addressing these cofactors has the potential to substantially improve survival for patients with advanced stage chronic pancreatitis.

## Supporting information

**S1 Appendix. NLP development processing.**
(DOCX)

**S1 Table. Supplementary tables for NLP processing and results.**
(DOCX)

**S1 Error analysis. Error analysis between the NLP algorithm and manual review against the validation dataset.**
(DOCX)

## Author Contributions

**Conceptualization:** Fagen Xie, Rex A. Parker, Bechien U. Wu.

**Data curation:** Fagen Xie, Yichen Zhou, Wansu Chen, Jemianne Bautista, Emilie T. Nguyen, Rex A. Parker, Bechien U. Wu.

**Formal analysis:** Fagen Xie, Qiaoling Chen, Yichen Zhou, Wansu Chen, Jemianne Bautista, Emilie T. Nguyen, Bechien U. Wu.

**Funding acquisition:** Bechien U. Wu.

**Investigation:** Qiaoling Chen, Yichen Zhou, Wansu Chen, Emilie T. Nguyen, Rex A. Parker, Bechien U. Wu.

**Methodology:** Fagen Xie, Qiaoling Chen, Yichen Zhou, Wansu Chen, Rex A. Parker, Bechien U. Wu.

**Resources:** Rex A. Parker, Bechien U. Wu.

**Software:** Qiaoling Chen, Yichen Zhou.

**Supervision:** Wansu Chen, Emilie T. Nguyen, Rex A. Parker.

**Validation:** Yichen Zhou, Jemianne Bautista, Emilie T. Nguyen, Rex A. Parker, Bechien U. Wu.

**Visualization:** Jemianne Bautista, Emilie T. Nguyen.

**Writing – original draft:** Fagen Xie, Wansu Chen, Bechien U. Wu.

**Writing – review & editing:** Qiaoling Chen, Yichen Zhou, Wansu Chen, Jemianne Bautista, Emilie T. Nguyen, Rex A. Parker, Bechien U. Wu.

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
