## [Decision Letter · Decision Letter 0]

29 May 2020

PONE-D-20-02468

Characterization of patients with advanced chronic pancreatitis based on electronic health data and high-throughput natural language processing of radiology reports.

PLOS ONE

Dear Dr. Wu,

Thank you for submitting your manuscript to PLOS ONE. After careful consideration, we feel that it has merit but does not fully meet PLOS ONE’s publication criteria as it currently stands. Therefore, we invite you to submit a revised version of the manuscript that addresses the points raised during the review process.

Please following reviewers' comments and address them point by point. 

We look forward to receiving your revised manuscript.

Kind regards,

Dejing Dou, Ph.D.

Academic Editor

PLOS ONE

2. Our internal editors have looked over your manuscript and determined that it is within the scope of our Digital Health Technology Call for Papers. This collection of papers is headed by a team of Guest Editors for PLOS ONE: Eun Kyoung Choe (University of Maryland, College Park), Chelsea Dobbins (University of Queensland), Sunghoon Ivan Lee (University of Massachusetts, Amherst), and Claudia Pagliari (University of Edinburgh).

The Collection will encompass a diverse range of research articles on digital health technologies ranging from technology design to patient care and health systems management.  Additional information can be found on our announcement page: https://collections.plos.org/s/digital-health-tech.

If you would like your manuscript to be considered for this collection, please let us know in your cover letter and we will ensure that your paper is treated as if you were responding to this call. If you would prefer to remove your manuscript from collection consideration, please specify this in the cover letter.

Reviewers' comments:

Reviewer's Responses to Questions

**Comments to the Author**

1. Is the manuscript technically sound, and do the data support the conclusions?

Reviewer #1: Yes

Reviewer #2: Yes

2. Has the statistical analysis been performed appropriately and rigorously? 

Reviewer #1: I Don't Know

Reviewer #2: Yes

3. Have the authors made all data underlying the findings in their manuscript fully available?

Reviewer #1: No

Reviewer #2: Yes

4. Is the manuscript presented in an intelligible fashion and written in standard English?

Reviewer #1: Yes

Reviewer #2: Yes

5. Review Comments to the Author

Reviewer #1: In this paper, the authors present a retrospective cohort study including patients diagnosed with chronic pancreatitis (CP). To extract specific features from free-text radiology reports, a natural language processing (NLP) approach was developed and manually evaluated using a training set and a validation set, respectively. The algorithm was then applied on a large sets of reports, to define a subset of patients with advanced disease vs. other patients. As a main result, patients with advanced CP were at increased risk for pancreatic cancer-related mortality. Excess all-cause mortality was driven primarily by underweight BMI, smoking and diabetes.

The use of NLP solutions to support retrospective research is an interesting topic, especially for those use-cases where relevant data is only contained in free text. Despite using a rule-based NLP approach, where concept variants might not be captured, evaluation results are promising. However, I believe there are a few points in the manuscript that could be extended or clarified.

1. In general, the section “Natural language processing for characterization of pancreatic imaging findings” would benefit from some clarification. Although more details are given in Appendix A, it would be helpful to briefly mention the high-level approach here, at least defining “search keywords”, “modifiers” and “exclusion terms” (especially considering that references to specific excluded terms are given). Were these keywords manually defined, or was there any data-driven step? Related to this, the authors refer to “further training of the computer-generated algorithm”. Was there any machine learning training involved?

2. Could the authors provide a few more details about the typical structure of documents? Do these usually contain predefined sections? How was spelling checking performed? More generally, do the authors think their approach could be easily reused or extended to process similar documents from a different institution?

3. I would suggest clarifying the section “NLP Performance evaluation” in Appendix A. Were results evaluated on a sentence level, report level, or patient level? This could be also clarified in the main manuscript. Also: could you please check the definitions of Specificity and NPV given here?

4. Did the authors perform any error analysis to identify particular issues with the developed NLP approach, e.g. how many false negatives were due to missing variants or misspellings?

5. The authors report p-values when comparing specific characteristics in advanced CP vs. other patients. Could they please clarify which tests were applied?

6. Table 3 reports causes of mortality in advanced CP vs. other patients. Could the authors clarify how these were defined? More specifically, is “Death from all causes” including all the causes listed below? If so, could you please check if reported sums are correct?

Minor comments

7. I would suggest adding acronyms (NLP and CP) the first time they are mentioned in the text.

8. Please check the numbers in the following sentence (Pg. 9) vs. Table 3: “Frequency of pancreatic cancer-related death was increased among patients with advanced CP (6.9% vs. 5.2%, p<.001)..."

Reviewer #2: The study aim in the abstract seems a bit too generic for the study, at least some more detail could be added here, like the focus on extracting the imaging features using NLP?

Validation cohort: was this cohort also manually reviewed for the imaging findings? This is not explicitly mentioned in the methods section, however, seems to be required?

It is a bit odd that the “Chronic Pancreatitis Study cohort” results paragraph comes second, whereas here the study cohort is presented? Any reason why this is not the first results paragraph?

For figure 2 is the advanced diseases based on NLP extracted radiographic findings or is it based on manually curated findings? It would be good to make this explicit.

Do the authors have any sense on how this algorithm would perform on external data from another hospital system? Are there any potential biases? It would be good to add this to the discussion.

How is this approach generalizable to new problems in other diseases areas, e.g. cirrhosis or NASH in livers. Based on the appendix, it does seem that this approach is difficult to generalize, and scale to other problems. Can the authors comment on this and add this to the discussion?

6. PLOS authors have the option to publish the peer review history of their article (what does this mean?). If published, this will include your full peer review and any attached files.

Reviewer #1: No

Reviewer #2: No

---

## [Author Response · Author response to Decision Letter 0]

26 Jun 2020

Response to comments

We have edited the revised manuscript to align with the PLOS ONE style requirements

2. Our internal editors have looked over your manuscript and determined that it is within the scope of our Digital Health Technology Call for Papers. This collection of papers is headed by a team of Guest Editors for PLOS ONE: Eun Kyoung Choe (University of Maryland, College Park), Chelsea Dobbins (University of Queensland), Sunghoon Ivan Lee (University of Massachusetts, Amherst), and Claudia Pagliari (University of Edinburgh).

The Collection will encompass a diverse range of research articles on digital health technologies ranging from technology design to patient care and health systems management. Additional information can be found on our announcement page: https://collections.plos.org/s/digital-health-tech.

If you would like your manuscript to be considered for this collection, please let us know in your cover letter and we will ensure that your paper is treated as if you were responding to this call. If you would prefer to remove your manuscript from collection consideration, please specify this in the cover letter.

We have indicated in our cover letter our desire for the manuscript to be considered for the Digital Health Technology collection.

 Kaiser Permanente Southern California (KPSC) institutional policy requires a Data Transfer Agreement to be completed naming all persons and entities that will have access to the data before any individual level data can be transmitted outside the organization. The KPSC Research & Evaluation Central Business Office handles data use agreements. For further inquiries: Judy Angmorter, senior contracts and grants administrator can be reached at Judy.J.Angmorter@kp.org

Updated captions and in-text citations have been included in the revised manuscript.

5. Review Comments to the Author

Reviewer #1: 

In this paper, the authors present a retrospective cohort study including patients diagnosed with chronic pancreatitis (CP). To extract specific features from free-text radiology reports, a natural language processing (NLP) approach was developed and manually evaluated using a training set and a validation set, respectively. The algorithm was then applied on a large sets of reports, to define a subset of patients with advanced disease vs. other patients. As a main result, patients with advanced CP were at increased risk for pancreatic cancer-related mortality. Excess all-cause mortality was driven primarily by underweight BMI, smoking and diabetes.

The use of NLP solutions to support retrospective research is an interesting topic, especially for those use-cases where relevant data is only contained in free text. Despite using a rule-based NLP approach, where concept variants might not be captured, evaluation results are promising. However, I believe there are a few points in the manuscript that could be extended or clarified.

Response: Thank the reviewer for the recognition on the contribution of our study and helpful suggestions. We addressed each point and incorporated them into the revision in detail (see below).

1. In general, the section “Natural language processing for characterization of pancreatic imaging findings” would benefit from some clarification. Although more details are given in Appendix A, it would be helpful to briefly mention the high-level approach here, at least defining “search keywords”, “modifiers” and “exclusion terms” (especially considering that references to specific excluded terms are given). Were these keywords manually defined, or was there any data-driven step? Related to this, the authors refer to “further training of the computer-generated algorithm”. Was there any machine learning training involved?

Response: We appreciate the Reviewer’s suggestions. Our NLP algorithm was a rule-based approach developed through an iterative process. The keywords were manually defined. As described in the main context, the search keywords for each feature were compiled based on the American Pancreas Association clinical guideline definitions, ontologies in the Unified Medical Language System and enriched from training datasets during the algorithm development to capture additional possible linguistic variations. We have expanded the description of these terms and the development process in the methods section of the revised manuscript (Page 4, lines 118-121, Page 5, lines 122-131). 

2. Could the authors provide a few more details about the typical structure of documents? Do these usually contain predefined sections? How was spelling checking performed? More generally, do the authors think their approach could be easily reused or extended to process similar documents from a different institution?

Response: In our care setting as in most clinical Radiology reporting, most of the imaging reports typically include sections of indication/clinical history, procedure, technique and findings as well as impression. Our NLP process only checked and corrected these mistyped, misspelled or concatenated words detected from our development training datasets. We added more detail descriptions regard structure of imaging reports and spell checking in the revision (Page 5, lines 134-135, Page 6, lines 154-155). 

Although our NLP algorithm was developed based on the imaging reports, the steps as we described in the Appendix A are generalizable and can be implemented in other care settings. It may yield some varied results due to the variation in format and presentation of reports, but the accuracy should not be significantly affected because the algorithm was not specified or limited to any fixed/strict formatted reports. We have added the relevant discussion in the Discussion section (Page 13, lines 322-326).

3. I would suggest clarifying the section “NLP Performance evaluation” in Appendix A. Were results evaluated on a sentence level, report level, or patient level? This could be also clarified in the main manuscript. Also: could you please check the definitions of Specificity and NPV given here?

Response: We thank the reviewer for their suggestions. The NLP performance was evaluated at the report level. We have clarified this in the main manuscript (Page 6, line 168) and Appendix A. We also have also clarified the definitions of Specificity and NPV in the revised Appendix A.

4. Did the authors perform any error analysis to identify particular issues with the developed NLP approach, e.g. how many false negatives were due to missing variants or misspellings?

Response: We did review the discrepancies between the NLP algorithm and manual review results for validation datasets. The details of error analysis was summarized in the S1 Error Analysis file and cited in the result section of the manuscript (Page 8, 221). 

5. The authors report p-values when comparing specific characteristics in advanced CP vs. other patients. Could they please clarify which tests were applied?

Response: To characterize differences in the clinical profile of patients with advanced CP we performed Chi-square test, t-test or Wilcoxon test to assess the age, sex and racial/ethnic distribution as well as the frequency distribution of etiologic risk factors. This information has been added to the revised Methods (page 7, lines 176-177).

6. Table 3 reports causes of mortality in advanced CP vs. other patients. Could the authors clarify how these were defined? More specifically, is “Death from all causes” including all the causes listed below? If so, could you please check if reported sums are correct?

Response: We thank the reviewer for identifying this discrepancy. We have identified the source of the error: when reporting individual causes of death, we failed to restrict the analysis to the end of the study period (2017). As a result, the sum of individual causes of death was greater than the total number of deaths during the study period, n=928. We have revised Table 3 and the manuscript text to reflect the updated numbers with respect to individual causes of death during the study period.

Minor comments

7. I would suggest adding acronyms (NLP and CP) the first time they are mentioned in the text.

Response: Thank the reviewer for the suggestion. We have added both acronyms at the first time place in the revision (Page 3, lines 86-87). 

8. Please check the numbers in the following sentence (Pg. 9) vs. Table 3: “Frequency of pancreatic cancer-related death was increased among patients with advanced CP (6.9% vs. 5.2%, p<.001)..."

Response: This has been corrected in the revised manuscript as the estimate should read (5.9% vs. 1.2%, p<.001).

Reviewer #2: 

The study aim in the abstract seems a bit too generic for the study, at least some more detail could be added here, like the focus on extracting the imaging features using NLP?

Response: Thank the reviewer for the good suggestions. We have added more details of our method in the abstract of the revision (Page 2).

Validation cohort: was this cohort also manually reviewed for the imaging findings? This is not explicitly mentioned in the methods section, however, seems to be required?

Response: Yes, the validation cohort was manually reviewed for the imaging findings. We have explicitly indicated this in the revised methods section of the updated manuscript (Page 6, line 148) 

It is a bit odd that the “Chronic Pancreatitis Study cohort” results paragraph comes second, whereas here the study cohort is presented? Any reason why this is not the first results paragraph?

Response: We have re-ordered the paragraphs as suggested in the revised manuscript. 

For figure 2 is the advanced diseases based on NLP extracted radiographic findings or is it based on manually curated findings? It would be good to make this explicit.

Response: Advanced disease was based on NLP extracted radiographic findings. This has been added to the figure legend for clarity.

Do the authors have any sense on how this algorithm would perform on external data from another hospital system? Are there any potential biases? It would be good to add this to the discussion.

Response: The steps of the NLP algorithm described in the Appendix A can be applied to other healthcare systems provided they have an electronic medical record system that includes free-text of radiology reports. When applied to other hospital systems, the accuracy for the established features should not be significantly affected by report formatting because the algorithm was not specified or limited to any fixed/strict formatted reports. We have added the relevant discussion in the revised Discussion section (Page 13, lines 322-326).

How is this approach generalizable to new problems in other diseases areas, e.g. cirrhosis or NASH in livers. Based on the appendix, it does seem that this approach is difficult to generalize, and scale to other problems. Can the authors comment on this and add this to the discussion?

Response: We thank the reviewer for raising this potential implication of the present study. Although our study focused on features relevant to outcomes in chronic pancreatitis, the approach and processing could be adapted to other medical conditions with relatively minor modifications, such as replacing the disease-specific keywords or terms of interest, corresponding modifiers and potential excluded terms. In addition, because our study focused on the pancreas, text descriptions related to other organ were excluded from the beginning of the processing as described in the Appendix A. However, this can be easily modified to include a specific organ of interest. We have added more discussion on this topic in the revised Discussion section (Page 13, lines 326-328).

---

## [Decision Letter · Decision Letter 1]

15 Jul 2020

Characterization of patients with advanced chronic pancreatitis using natural language processing of radiology reports.

PONE-D-20-02468R1

Dear Dr. Wu,

We’re pleased to inform you that your manuscript has been judged scientifically suitable for publication and will be formally accepted for publication once it meets all outstanding technical requirements.

Kind regards,

Dejing Dou, Ph.D.

Academic Editor

PLOS ONE

Additional Editor Comments (optional):

Reviewers' comments:

Reviewer's Responses to Questions

**Comments to the Author**

1. If the authors have adequately addressed your comments raised in a previous round of review and you feel that this manuscript is now acceptable for publication, you may indicate that here to bypass the “Comments to the Author” section, enter your conflict of interest statement in the “Confidential to Editor” section, and submit your "Accept" recommendation.

Reviewer #1: All comments have been addressed

Reviewer #3: All comments have been addressed

2. Is the manuscript technically sound, and do the data support the conclusions?

Reviewer #1: Yes

Reviewer #3: Yes

3. Has the statistical analysis been performed appropriately and rigorously? 

Reviewer #1: Yes

Reviewer #3: Yes

4. Have the authors made all data underlying the findings in their manuscript fully available?

Reviewer #1: No

Reviewer #3: No

5. Is the manuscript presented in an intelligible fashion and written in standard English?

Reviewer #1: Yes

Reviewer #3: Yes

6. Review Comments to the Author

Reviewer #1: (No Response)

Reviewer #3: This paper develops a NLP algorithm to identify five specific features of patients with chronic pancreatitis from their radiology reports, and identify a subset of patients with advanced CP based on these features. The proposed method can be applied to other data sources with text and other medical conditions after modification.

7. PLOS authors have the option to publish the peer review history of their article (what does this mean?). If published, this will include your full peer review and any attached files.

Reviewer #1: No

Reviewer #3: No

---

## [Editor Report · Acceptance letter]

7 Aug 2020

PONE-D-20-02468R1 

Characterization of patients with advanced chronic pancreatitis using natural language processing of radiology reports. 

Dear Dr. Wu:

I'm pleased to inform you that your manuscript has been deemed suitable for publication in PLOS ONE. Congratulations! Your manuscript is now with our production department. 

Kind regards, 

on behalf of

Professor Dejing Dou 

Academic Editor

PLOS ONE